# Closed-Loop Recycling and Remanufacturing of Polymeric Aircraft Parts

Marko Hyvärinen *, Mikko Pylkkö and Timo Kärki

Fiber Composite Laboratory, Lappeenranta–Lahti University of Technology LUT, P.O. Box 20,
53850 Lappeenranta, Finland
* Correspondence: marko.hyvarinen@lut.fi

**Abstract:** The aviation industry is facing the challenge of reducing fossil fuels and emissions. Fuel efficiency is improved by making efficient powerplant systems and lighter aircraft. Modern passenger aircraft utilize polymeric and polymeric composite materials to achieve lighter structures without compromising strength. The European Union already has legislation to prevent landfilling and to increase the use of recyclable materials in the automotive industry. While older-generation aircraft, made mainly from metallic materials, are easily dismantled and recycled into other uses, such a process does not yet exist for aircraft made from composite materials. In the coming years, the industry will have to answer the question of how retired polymeric composite aircraft structures are to be recycled. One solution to increase the life cycle of polymeric and polymeric composite parts would be closed-loop recycling. In this paper, a perspective of the closed-loop recycling of polymeric aircraft parts is discussed. The technical aspects of recyclability and the different business models for the remanufacture of a finger pinch shroud certified for use in Airbus A350-900 passenger aircraft are investigated. The results show that closed-loop recycling is possible for polymeric aircraft parts. Future studies could include studying an LCA between virgin and recycled materials for a certain part.

**Keywords:** closed-loop; recycling; manufacturing; aircraft; business model

## 1. Introduction

Fuel is one of the largest expenses for airline operators due to increasing oil prices. In conjunction with a growing awareness about climate change, it has forced airline operators, together with manufacturers, to seek ways to reduce fuel consumption and emissions. Manufacturers have been tackling this problem by improving the technology and efficiency of engines, making it possible to reduce the number of engines on an aircraft, and, recently, by making aircraft structures lighter. Lighter structures require less thrust and, thus, less fuel [1].

The development of highly engineered materials such as fiber-reinforced polymers (FRP) has helped manufacturers in the search for lighter aircraft. Such materials are generally also called composite materials, and are desirable for aircraft manufacturers due to their excellent mechanical properties and low weight [2]. These properties enable the production of lighter structures, making lighter aircraft, leading to decreased fuel consumption. However, composite materials possess a fundamental problem in that their recyclability is currently poor. In addition, producing these materials requires significant energy and it increases emissions [3].

The evolution of FRP and the development of their processability has enabled the introduction of composite materials in the primary structures of aircraft [4]—the fuselage skin, structures, wings, and cabins [5]. In addition to primary structures, polymeric and composite materials are heavily used in aircraft interior applications. While cabin items may not be required to withstand high structural loads, they must be able to resist passenger use,

and in addition are regulated with tight fire, smoke, and toxicity requirements. Typically, applications for aircraft interiors are manufactured using thermoplastic or reinforced thermoset materials. Large wall and ceiling panels, seat shells, and monuments usually have a honeycomb structure reinforced with aramid or carbon fibers. These applications include floor and ceiling panels, seats, stowage, lavatories, galleys, and different kinds of dividers [6,7].

Currently, neither legislation nor aviation regulations address aircraft-related polymeric or polymeric composite waste materials. In the past 35 years, around 16,000 aircraft have been retired. The International Civil Aviation Organization (ICAO) estimated that, in the next 10 years, there will be an additional 11,000 aircraft removed from service. Due to the lack of end-of-life regulation and the estimated number of aircraft to be retired, the ICAO has published manuals to support airline owners and operators in dismantling and recycling end-of-life aircraft [8].

The world's leading passenger aircraft manufacturers, namely, Airbus and Boeing, have also researched the recyclability of aircraft in their end-of-life phase. Airbus conducted a full-scale experiment on aircraft recycling and the results revealed that 85 wt% of passenger aircraft components could be recycled [9]. Boeing has tested the use of carbon fiber recycled from its 777 and 787 aircraft production lines. The company is already viewing it as one end-of-life option. A cabin sidewall made of these recycled materials is already in use in one 737 MAX aircraft [10].

The difference in recyclability between older-generation aircraft and modern aircraft with more composite materials is significant. In 2021, the Finnish airline operator, Finnair, and its affiliated company, Finnair Technical Operations, together with the recycling company, Kuusakoski Oy, dismantled and recycled one of the operator's oldest Airbus A319 passenger aircraft. The aircraft had entered service around the year 2000 and was retired at the age of 21 years. Of the aircraft, 49.1% was recycled, with most of the material being aluminum—about 15 tons. Additionally, 38.5% of the aircraft was reused in the form of components and parts that could be used as spare parts for the rest of the fleet. Moreover, 7.4% of the aircraft was recovered as energy, and 0.8% of the aircraft was disposed of. Finally, 4.2% of the aircraft was used for research, focusing on the utilization of composite materials [11]. The research results are not yet available.

Choosing the most suitable recycling method is also essential for recycling plastics, and alternative methods include mechanical and chemical recycling [12]. New recycling technologies such as thermomechanical recycling and biological depolymerization are also being developed [13]. Mechanical recycling can only be applied to linear or loosely crosslinked polymer components, such as polypropylene (PP), polyethylene (PE) and polystyrene (PS). Whereas for polymethylmethacrylate (PMMA), which is a widely used polymer in the aerospace and automotive industries, e.g., for windshields and windows, mechanical recycling is less desirable because the resulting product does not meet the desired optical properties [14]. Additionally, it has been reported [15] that the chemical recycling of PMMA can further reduce the environmental impact when mechanical recycling has limited potential.

In this paper, the closed-loop recycling of polymeric aircraft parts and the manufacture of new equivalent parts from recycled materials is discussed. The recycling of aircraft is mainly studied from an economic perspective, e.g., as reported in [16,17]. Business models for end-of-life treatment have also been discussed, e.g., as highlighted in [18,19], but business models for the recycling and remanufacturing of polymeric aircraft part have not been presented. A passenger aircraft part, i.e., a polymeric finger pinch shroud from a Safran Seats US Z300 passenger seat certified for use, e.g., in Airbus A350 aircraft, and approved by the European Union Aviation Safety Agency (EASA) and the Federal Aviation Administration (FAA), was used as an example product in this study. The finger pinch shroud is made of virgin polypropylene (PP). The part is installed to the hinge of the arm rest of a seat to prevent passengers from accidentally placing their fingers into the hinge. The finger pinch shroud was chosen as a product example because of its simple

geometry, which also makes measuring, reverse engineering, and 3D printing easy. The part has already been superseded by the Vendor Service Bulletin due to its tendency to break away from the notch area. The studied aircraft part is presented in Figure 1. This study aims to discuss the technical aspects, limitations, workflow, and business models for the closed-loop recycling of polymeric aircraft parts.

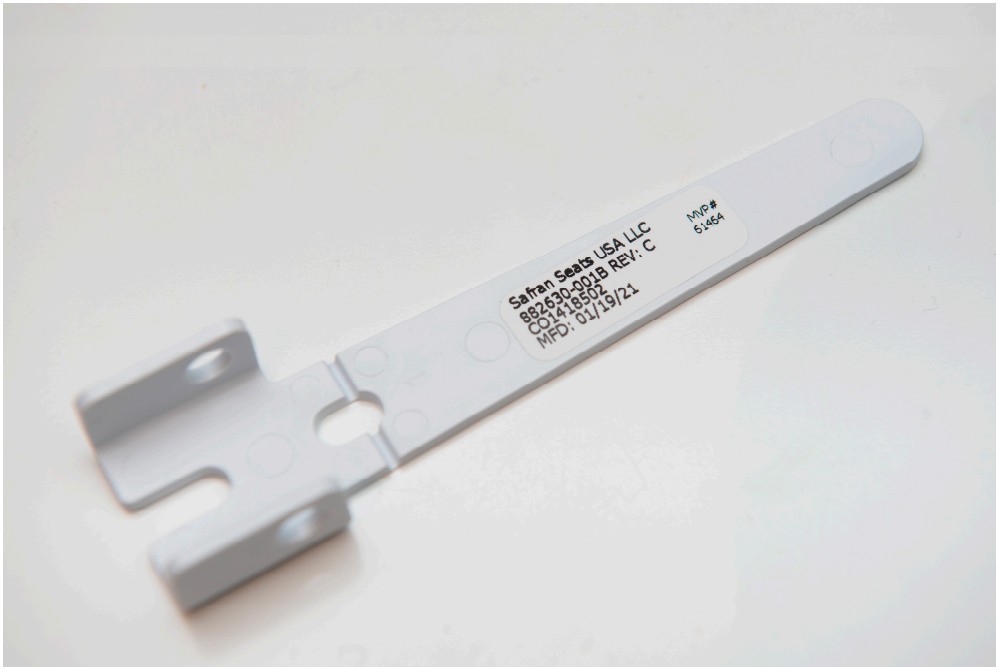

**Figure 1.** The studied aircraft part, namely, a finger pinch shroud from a Safran Seats US Z300 passenger seat.

## 2. Closed-Loop Recycling in Aviation

The ISO 14044 standard makes a distinction between two types of recycling: closed-loop recycling, which is a process whereby a product is recycled into a new, similar product without changing the properties of the recycled material; and open-loop recycling, which is a process where a material may be recycled into other product systems, and the recycled material properties may be altered [20].

Closed-loop recycling shall not be mixed with remanufacturing. In remanufacturing, a used part is retained when making a new part, and it retains its general form. In recycling, the part is broken down to the component level, through melting, crushing, or reprocessing, and then remanufactured into a new part. Remanufacturing often requires the complete disassembly, inspection, measuring, and cleaning of the old part before it can be reused in a new product. A remanufactured part is given the same warranty as a new part. Remanufacturing helps to save raw materials and energy, as well as to reduce emissions [21–23]. In aviation, remanufacturing is referred to by the term, overhaul, which is usually undertaken for life-limited parts such as engines, landing gears, and other components. It is a crucial part of business for the original equipment manufacturer (OEM) and maintenance, repair, and overhaul (MRO) companies in the aviation industry.

Closed-loop recycling in aviation has been studied by Hashemi et al. who discussed the use of a closed-loop supply chain and remanufacturing as a part of the manufacturing of new components for aviation and provided a mathematical model of a closed-loop supply chain. These studies were mainly related to remanufacturing, while closed-loop recycling methods or materials were not analyzed [24,25].

Despite efforts to enable a circular economy in the aviation industry, currently, only a small proportion of aircraft and their parts are recycled, and this typically occurs after the aircraft is removed from service. In many cases, an aircraft is retired due to increasing

maintenance costs, even if it has operational life left. Currently, most recycled aerospace materials are delivered to other industries or waste companies or are landfilled. No major aerospace company enables direct recycling within the industry yet. It is predicted that, if such a business existed, it would increase the value of the recycled material and reduce the need for virgin materials [26].

Key questions related to the implementation of closed-loop recycling in the aviation industry include, for example, how to certify parts made from recycled materials and whether recycling would be economically viable. To use closed-loop recycling for the manufacturing of new parts, material properties, in particular flammability, must be proven to meet the regulatory requirements. Would it be cost-effective to use only the recycled material, or would it be possible to mix it with a virgin material? In both cases, the certification of the material must be performed carefully. In addition, would the closed-loop recycling offer economic benefits to manufacturers, for example, in the form of lower material costs? Another issue to consider is whether additive manufacturing would be the best approach to manufacturing or whether injection molding would be a more practical manufacturing technology.

A great challenge in the practical implementation of closed-loop recycling is the verification and certification of the material of a part or product. The use of closed-loop recycling in the manufacture of parts would offer certain opportunities, one of which is the possibility of producing parts on-demand. To certify and manufacture a part on-demand, there must be confidence that the material will always have the same properties, regardless of its origin. EASA produces documents to guide design offices in implementing EU legislation in the design process, and the certification of parts and structures for large aircraft is guided by the CS-25 Large Aeroplanes Certification specification.

The certification of the material and the manufactured part ensures that the part is airworthy and can be installed on an aircraft. The certification is a detailed process and any justification for each paragraph in CS-25 must be substantiated by test data or certificates provided by the material producer. When the closed-loop recycling process for OEM parts is observed, such data may not be available. In such a case, the recycling and manufacturing operator should establish a process to identify the material properties such as strength and flammability.

An operator could, for example, gather the recycled materials, perform material identification for the materials using, e.g., NIR spectroscopy, and sort the materials based on their properties. Subsequently, the material could be processed into granules from which test samples for strength, fire, and flammability testing could be prepared. If all the tests are passed, the material might be fit for the intended use and could be certified. If one or more tests fail, then the operator must consider adding some compounds or additives to the material to improve the desired properties. This kind of process would enable on-demand manufacturing.

Another approach to certification would be to perform the material identification and sorting as described above and then to manufacture additional parts that are then tested accordingly. A problem with this kind of approach is that every time a part was required to be manufactured on-demand, a test part would have to be manufactured and tested. This would slow down the process and possibly increase the costs as well.

For example, the automotive industry is already utilizing recycled materials, mainly for interior parts or parts not directly affecting driving safety. In the year 2000, the European Union (EU) set binding legislation for the automotive industry to reduce waste arising from end-of-life vehicles. According to this EU directive, the reusability and recyclability of the materials used in vehicles manufactured from the beginning of 2015 must be at least 95% of the vehicle's average weight [27]. Due to this legislation, car manufacturers have been looking into using recycled materials, and some have recycled materials in place in their production models. The automotive industry is, therefore, a valid benchmark for the aerospace industry.

While the economic perspective must be evaluated when studying the use of alternative materials, the environmental aspects are equally as important, if not the main priority. The use of leftover materials in the manufacturing of new products certainly offers cost benefits, but for a large industry such as aviation, it is crucial to be able to decrease the emissions and energy consumption in manufacturing and operations. This has already been recognized in the automotive industry. For example, BMW has stated that the development of electronic vehicles cannot rely only on primary materials, as the use of recycled materials helps to reduce $CO_2$ emissions and enables the same materials to be used for longer [28].

In terms of manufacturing methods, Ford has studied the use of additive manufacturing for production waste in injection-molded components, and some of these experiments have already been implemented in production. For example, the Ford Super Duty F-250 already has fuel-line clips made from powder waste and old printed parts. The 3D printing waste, powder, and old parts come from a company working in the dental sector. According to Ford, the clips made using this method are 7% lighter and cost 10% less than clips made using traditional methods [29]. Audi has also used recycled materials to print tools and jigs for manufacturing. The materials come from its own packaging waste, which is then used to make filament. Filament is used to 3D print tools such as pushrods and jigs to align Audi logos [30].

Tightening environmental legislation will force different industrial sectors to improve and develop material efficiency and waste recycling. In the automotive sector, there have long been regulations on the reuse and recycling of materials from end-of-life vehicles. In the coming years, similar regulations are expected for the aircraft industry. This will also force the aircraft industry to improve its practices towards better material efficiency and recycling.

## 3. Business Models

Currently, the use of recycled materials for aircraft parts is a relatively new idea. Cabin-related parts would be the first applications for such materials due to the looser regulatory requirements. Even though parts in the cabin must fulfill several requirements related to their strength, fire, and flammability properties, it is easier to tackle those requirements before thinking of implementing such materials into primary structures or powerplant-related parts.

It is worthwhile to consider whose responsibility the closed-loop recycling of an aircraft part would be. Three different operators who are needed to run this kind of business can be recognized:

1. Aircraft manufacturers or original equipment manufacturers
2. Third-party manufacturers
3. Airlines (technical departments).

The abovementioned operators would have the resources for design and manufacturing, and already possess the regulatory qualifications to carry out the design, manufacturing, and certification of the parts. Small airline operators do not necessarily possess these qualifications but, in this analysis, it is assumed that the airline operator has its own technical department with EASA or FAA qualifications for the design and manufacture of parts.

The aircraft manufacturers and original equipment manufacturers who are listed here together as aircraft manufacturers, e.g., Airbus, Boeing, and Embraer, mostly do not manufacture cabin products themselves, but prepare the guidelines for specifications. The manufacturing of products is usually contracted out to a certain specialized manufacturer—for seats, for example, to Recaro, Safran Seats US, or B/E Aerospace. When ordering a new aircraft, the airline operator may choose seats from the aircraft manufacturer's catalogue, and sometimes, the airline wants to specify and tailor their own design for seats, but usually the aircraft manufacturer is involved in this process as well.

A company specialized in manufacturing, for example of polymeric or composite products, may already have the required machinery and design capability, but not the expensive and difficult qualifications. Without a customer base in the field of aviation, establishing such regulatory qualifications can require a large investment. Once qualified, they are counted as a third-party manufacturer, unless they offer a product for which they would be counted as the original equipment manufacturer.

### 3.1. Aircraft Manufacturer or Original Equipment Manufacturer

The motivation of an aircraft manufacturer or original equipment manufacturer (OEM) to participate in closed-loop recycling could be related to possible future regulations or the preparation for them; cost reduction, assuming recycled materials would be cheaper than virgin materials; a reduction in emissions and energy consumption; and brand image. An OEM already has the sourcing capability for virgin materials and might already be able to supply recycled materials as well. If not, the OEM's procurement department could have the capability for sourcing recycled material providers. Sometimes, large aircraft manufacturers even buy back their older model aircraft when selling newer models to a customer. From these older, almost retired aircraft, the OEM could source the desired materials and recycle them into new parts and products.

Another model of sourcing for an OEM could be, for example, establishing an agreement with an airline operator, whereby they provide damaged and scrapped parts for the recycling and manufacturing of new parts for a discounted price. In this way, the OEM could get raw materials for manufacturing at a cheaper price and, at the same time, produce new parts with lower emissions. Table 1 provides a SWOT analysis of an OEM performing closed-loop recycling.

**Table 1.** SWOT analysis of an OEM manufacturer of recycled material parts.

| Strengths | Weaknesses |
|---|---|
| • Regulatory qualifications to carry out design, manufacturing, and certification.<br>• Research and development (R&D) capability.<br>• Human resources (design and manufacturing personnel, and certifiers). | • A completely new approach to manufacturing high-tech products.<br>• No experience in the recycling and recovery of materials. |
| **Opportunities** | **Threats** |
| • The possibility of reducing emissions and energy consumption during manufacturing.<br>• The possibility of reducing the use of fossil fuels.<br>• Sourcing of materials through warranty programs or buyback agreements.<br>• Competitive advantage from the use of sustainable materials.<br>• Improving brand image. | • Customers' prejudice towards the use of recycled materials instead of new, high-tech materials. |

The SWOT analysis shows that there are multiple strengths and opportunities an aircraft manufacturer/OEM possesses. The strengths would enable starting such a process and the opportunities would support the operation. The recognized threats show that the recycled material might not appear trustworthy or high-tech to passengers but, given the fact that the certification and manufacturing process is tightly regulated, manufacturers would be able to tackle these possible negative assumptions. Additionally, the manufacturer would benefit immensely from being able to advertise the usage of recycled materials. An airline would also benefit from being able to advertise using sustainable materials on their fleet and this could be a selling point.

An aircraft manufacturer/OEM would possess all the required resources for starting closed-loop recycling, and the opportunities for them would be significant. There are threats as well, but nothing that could not be overcome.

### 3.2. Third-Party Manufacturer

Closed-loop recycling could also be performed by a third-party operator, and such an operator should have, for example, EASA or FAA design organization and manufacturing organization approvals. Organizations operating under the FAA are usually called parts manufacturer approval (PMA) manufacturers, whose aircraft parts are certified and approved by the FAA [31]. A PMA part does not require OEM approval, and such parts can usually compete with pricing and design improvements over comparable OEM parts. PMA parts have a large market, especially in the United States. These parts can be used outside the United States as well, for example, the EASA has guidance and approval for using these parts. European design and manufacturing organizations usually act under EASA Part-21 approvals. The Part-21 regulation lays down common technical requirements and administrative procedures for the airworthiness and environmental certification of products, parts, and appliances [32].

A third-party manufacturer might be in a position where they can compete with aggressive pricing. In this case, they could also set up a process where the customer provides them with old, unserviceable parts to be reproduced as new parts. Such an organization could also purchase larger old assemblies (e.g., seats, lavatories, and other cabin modules where the desired materials are used) or even aircraft to recover the materials and reproduce them into the desired parts and products. Table 2 presents a SWOT analysis of a third-party manufacturer.

**Table 2.** SWOT analysis of a third-party manufacturer of parts made from recycled materials.

| Strengths | Weaknesses |
|---|---|
| • Regulatory qualifications to carry out design, manufacturing, and certification.<br>• Human resources (design and manufacturing personnel, and certifiers). | • A completely new approach to manufacturing high-tech products.<br>• No experience in the recycling and recovery of materials.<br>• Sourcing of materials. |
| **Opportunities** | **Threats** |
| • The possibility of reducing emissions and energy consumption during manufacturing.<br>• The possibility of reducing the use of fossil fuels.<br>• Competitive advantage from the use of sustainable materials.<br>• Improving brand image. | • Airline restrictions on the use of third-party-manufactured parts.<br>• Customers' prejudice towards the use of recycled materials instead of new, high-tech materials. |

A third-party manufacturer would already possess certain strengths such as the resources for performing the design, manufacturing, and certification. The opportunities for them would include an improvement to brand image (e.g., airlines could benefit from buying more sustainable spare parts), as well as the possibility of reducing emissions, energy consumption, and the need for the use of fossil fuels. Using closed-loop recycling might also be a benefit to them over other manufacturers, in addition to improving their brand image. The weaknesses include a lack of knowledge about working with such materials and possible problems related to materials sourcing. A threat for a third-party provider is that many airlines are leasing whole fleets, or part of them, and the lease agreements may restrict the airlines from using third-party parts. Additionally, customers' negative presumptions related to the use of recycled materials can be considered a threat.

While a third-party provider would already have the necessary qualifications and resources to carry out closed-loop recycling, they have a limited customer base and a somewhat limited scope of parts. The third-party provider could benefit from the sustainability aspect in marketing, but would not be able to provide airlines with as great an advantage as the major manufacturers, especially aircraft manufacturers, could. This brand image advantage could only be used in business-to-business relationships, which are usually guided by the economy.

### 3.3. Airlines

Exploiting closed-loop recycling might be of interest for an airline as well. An airline would already have access to materials via the unserviceable and scrapped parts removed from their fleet, which makes them a likely operator in the field; however, this kind of operation would require the airline having decent technical operations resources, and especially EASA or FAA design organization and manufacturing certifications. Without such approvals, the airline would have to hire a subcontractor to carry out the design and manufacturing; the subcontractor would in this case be an OEM, a PMA manufacturer, or an MRO organization.

An airline's motivation to start closed-loop recycling might also be related to lowering emissions and maintenance-related costs. OEM pricing for parts is usually high, and by utilizing its own manufacturing capabilities an airline could pursue cost savings. At the same time, the need for stock management would decrease. This would also lower emissions as no shipping of the parts would be required. One major advantage for an airline carrying out closed-loop recycling would be on-demand manufacturing, where in theory a damaged part could be recovered into a new part within a relatively short time. Table 3 presents a SWOT analysis for an airline operator performing closed-loop recycling.

**Table 3.** SWOT analysis for an airline technical operation department to run closed-loop recycling.

| Strengths | Weaknesses |
|---|---|
| • Regulatory qualifications to carry out design, manufacturing, and certification.<br>• Human resources (design and manufacturing personnel, and certifiers). | • A completely new approach to manufacturing high-tech products.<br>• No experience in the recycling and recovery of materials.<br>• Possible investments required for material tests, recovery, and manufacturing. |
| **Opportunities** | **Threats** |
| • Cost savings from materials and spare parts.<br>• Manufacturing on-demand.<br>• The possibility of reducing storage and handling costs.<br>• The possibility of reducing emissions and energy consumption during manufacturing.<br>• The possibility of reducing the use of fossil fuels.<br>• Competitive advantage from the use of sustainable materials.<br>• Improving brand image. | • Possible lack of knowledge and experience in the manufacture of polymeric parts. |

In this SWOT analysis, it is presumed that the airline would have EASA- or FAA-approved Part-21 design and manufacturing organizations in place. This would give them the strength of having the approval to certify the design and manufacturing, and would also mean that they have the resources to carry out such tasks. The opportunities for an airline in the very competitive market would give them a certain brand image and sustainability-related assets such as the possibility of reducing emissions and the use of fossil fuels. It could also offer them the possibility of on-demand manufacturing and, in this way, decrease stock handling costs. The value of on-demand manufacturing cannot be underestimated as in certain situations, unserviceable parts may lead to the blocking of a

seat, which would lead to economic losses. This is an undesirable situation as airlines tend to make every seat available as much as possible. In such cases, the damaged part could be manufactured within a short time period and installed in the aircraft, thereby avoiding the possible blocking of a seat. Weaknesses for an airline might be that this is a completely new territory where they might not have been operating before, and that possible investments would have to be made. Threats include a lack of knowledge of working with polymeric parts and how this would affect design and manufacturing.

Airlines could benefit from the sustainability-related brand image that closed-loop recycling would provide. They would also benefit from the possibility of utilizing the materials from broken and unserviceable parts. On-demand manufacturing would help reduce stock costs and could help with obtaining spare parts on short notice. On the other hand, such an operation is by no means a core business of an airline and would require significant investment. If an airline considers that the investment is justified by the potential for brand image and potential cost savings, it would be worthwhile to set up this type of operation. Figure 2 shows an outline for a process description of an airline running closed-loop recycling.

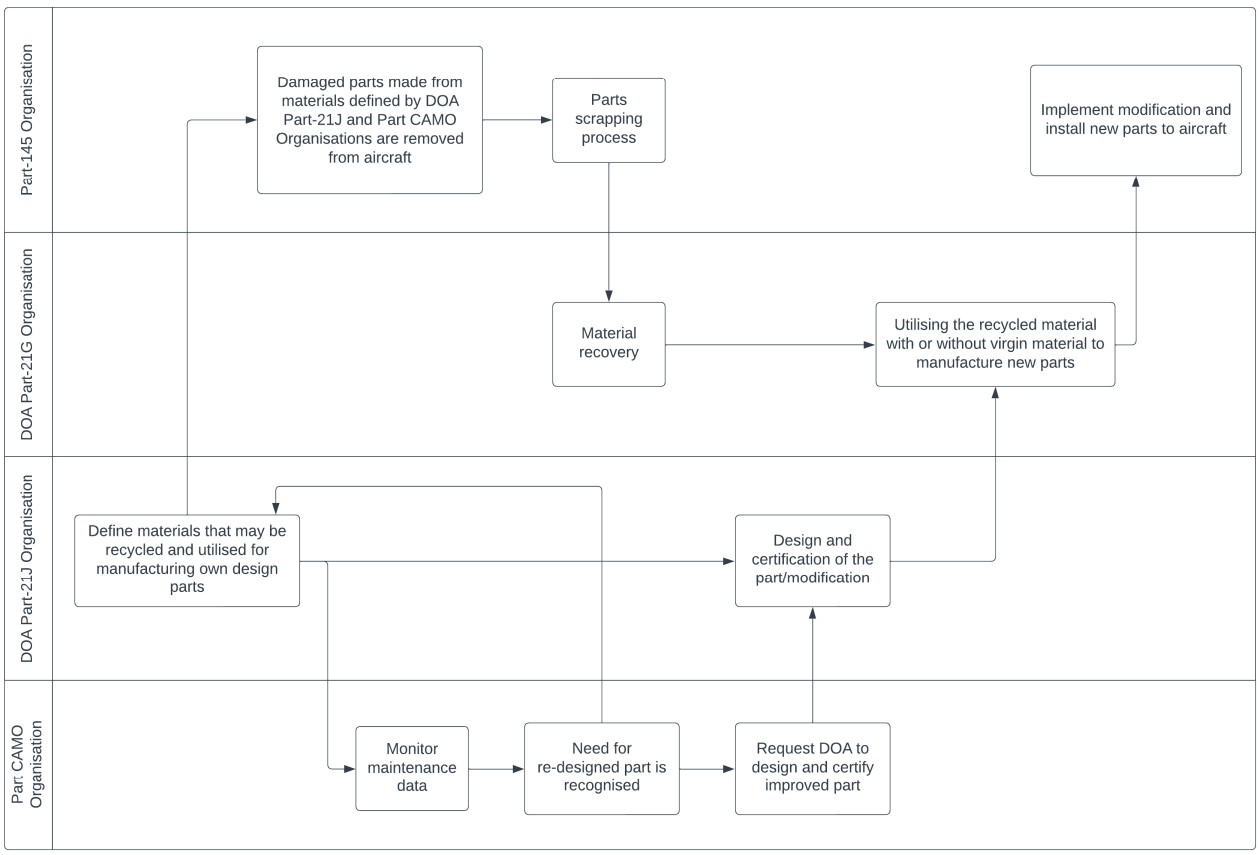

**Figure 2.** An outline for process description of an airline running closed-loop recycling.

For an airline to run this kind of operation, it should already have an EASA- or FAA-approved Part-145 (aircraft maintenance organization), Part-CAMO (continuous airworthiness management organization), Part-21J (design organization), and Part-21G (manufacturing organization). The process could be initiated by the Part-CAMO organization, which would recognize the need for redesigned parts based on maintenance data. Together with a DOA (design organization approval) Part-21J organization, the suitable materials, and parts where these kinds of materials are used, could be defined. A Part-145 organization would remove such parts when they are found to be damaged and scrap them according to the regulatory requirements. The scrapped parts would be supplied to a Part-21G organization that would recover the material from the parts and manufacture

new parts using these materials based on design data supplied by Part-21J. Part-145 would then modify and install the new parts in the aircraft. The process diagram for this kind of operation is shown on the next page.

Of these three operators, an aircraft manufacturer or OEM would have the greatest strengths and strongest opportunities to start closed-loop recycling. They would already possess research and development (R&D), design, procurement, and manufacturing organizations. They would benefit the most from the positive brand image in passengers' eyes, as well as using it as a competitive advantage when selling aircraft to airlines. While an airline would also have certain reasonable benefits from closed-loop recycling, setting up such an operation might be a big task and, in the end, it would not be their core business. A third-party manufacturer could profit from the sustainable brand image, but their audience would be restricted and would usually tend to choose them over an OEM only if it were economically viable.

## 4. Manufacturing Method

The manufacturing process itself is a topic worthy of separate discussion. Although additive manufacturing (AM) techniques have certain advantages over injection molding, which is a standard for the mass production of polymer and polymer composite parts, the question remains of whether additive manufacturing is the best solution for the closed-loop manufacturing of recycled parts. If environmental considerations are the driving force for seeking to utilize recycled materials, sustainable manufacturing methods should be chosen that support a life cycle assessment (LCA) of the product.

When choosing a manufacturing method, the constraints and potential of the method must be assessed. Comparative research has been carried out between additive manufacturing and injection molding in relation to manufacturing methods and surface quality. The conclusions of these studies show that the part may require redesigning, depending on the manufacturing method, to achieve the best outcome.

In some cases, the material composition may need to be evaluated to determine whether some additives should be used with the material to achieve similar results [33]. It was also found that there was almost no difference in the surface finish, fracture surfaces, or mechanical properties between parts made by additive manufacturing and injection molding [34].

When using injection molding, the possible accumulation and buildup of oil and certain other nanoparticles must be considered. The number of times the material can be recycled before such additives begin to affect the properties of a material must also be considered. A similar effect would not be seen with additive manufacturing techniques, as no oils or other additives are used in the process, and this is worth investigating.

For the mass production of small and medium-sized polymer and polymer composite products, injection molding is often the best manufacturing method. Injection molding can be used to manufacture products from a variety of thermoplastic materials, but it is expensive if batches are small. The comparison can also be made from an investment perspective. Several studies have discussed the economics and cost models of additive manufacturing [35–37]. Energy consumption has also been compared between AM and conventional manufacturing methods, such as injection molding, in several studies [38–40]. Additive manufacturing machines are generally cheaper than injection-molding machines and, in addition, molding requires individual molds for each part to be manufactured, as well as a much longer setup time. For small batches, e.g., a few hundred units, additive manufacturing may be the more reasonable choice [35].

For an operator manufacturing smaller batches or manufacturing only on-demand (for example, small batches of highly customized parts, or an airline that wants to produce spare parts with on-demand manufacturing), additive manufacturing may be the best option. Injection molding loses the flexibility, speed, and revisioning possibilities of additive manufacturing, but it makes more sense to use injection molding if one needs to produce

large batches. For an operator manufacturing large batches, injection molding would be the best choice as a manufacturing method.

## 5. Conclusions

In the future, the number of aircraft retiring will increase. At the same time, the regulations on aircraft recycling will also become tighter. Composite materials account for more than 50% of the weight of modern aircraft; therefore, the share of composite materials in the recycling of retired aircraft can be expected to increase in the future. One of today's challenges is that there is currently no technically satisfactory recycling technology for composites. The aviation industry aims to find a model for a comprehensive recycling process to recycle aircraft in an environmentally friendly and cost-effective way as well as to use recycled parts or materials in variety of applications [16,41].

The closed-loop recycling of a polymeric finger pinch shroud from a Safran Seats US Z300 passenger seat certified for use on the Airbus A350 aircraft was investigated. The methods used in this work have proved that the recycling and reproducing of such a part, or a part designed for a similar use using the same material, is feasible.

As the economic advantage cannot be shown clearly within the scope of this work, the possible business model must be evaluated according to other applicable attributes. The business model of closed-loop recycling studied herein would be most suitable for either aircraft or original equipment manufacturers. They would already possess capable R&D, design, procurement, and manufacturing organizations. A positive sustainability-related brand image would also give them a great advantage over other competitors and raise the value of their brand in the eyes of the passengers of the airlines.

Future studies could include studying the reinforcement of recycled materials, the use of recycled materials in other more critical structures of an aircraft, making an LCA between virgin and recycled materials for a certain part, and elaborating on a description of the actual recycling process.

**Author Contributions:** Conceptualization, M.H. and M.P.; methodology, M.H., M.P. and T.K.; writing—original draft preparation, M.H. and M.P.; writing—review and editing, M.H. and M.P.; visualization, M.P.; supervision, T.K.; project administration, T.K. All authors have read and agreed to the published version of the manuscript.

**Funding:** This research received no external funding.

**Data Availability Statement:** All data generated or analyzed during this report are included in this published article.

**Conflicts of Interest:** The authors declare no conflict of interest.

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
