# Peer review of "Closed-Loop Recycling and Remanufacturing of Polymeric Aircraft Parts"

_jcs, doi:10.3390/jcs7030121_

Round 1
Reviewer 1 Report
In this paper, several aspects of the closed-loop recycling of polymeric parts of aircrafts is reviewed. The focus of the paper is on aspects such as certification and SWOT analyses for different stakeholders. As such, the paper is an interesting contribution. I suggest that the comments below are addressed before publication.
- I understand that the focus of the paper is not on the actual closed-loop recycling processes. Nevertheless, I suggest to (briefly) situate the relevant recycling techniques with respect to e.g. mechanical/chemical recycling.
- PMMA is an important polymer in the aviation/automobile sector with interesting possibilities for recycling (see e.g. Polymers 2020, 12(8), 1667). I suggest to highlight this.
- The link between the finger pinch shroud (Figure 1) and the remainder of the review is quite unclear. I think the abstract and conclusions focusing on this specific part should be adjusted.
- The title of Section 4 should be corrected.
Author Response
Dear reviewer,
Thank you for your valuable comments. I have revised the article. The revisions can be seen in the new version of the article.
Reviewer 2 Report
In the Manuscript entitled “Closed-loop recycling and remanufacturing of polymeric aircraft parts”, the authors investigated the closed-loop recycling of polymeric aircraft parts. They found that airbus passenger aircraft was tested for its material and strength properties of the specimen and reproduced using additive manufacturing. They also provided circular economy and business models for these End-of-Life aircraft parts.
Overall, the research article has many shortcomings, which should be addressed before possible publication in JCS journal.
Please find the attached annotated file to see my comments.
Journal of Composites Science journal publishes high-quality research articles related to recycling of composites. Based on my comments, the recommendation is Major Revision.

Author Response

(The authors gave the same response as above.)

Round 2
Reviewer 1 Report
The comments have been addressed.
Reviewer 2 Report
Accepted